# Evaluation of Nutri-Score in Relation to Dietary Guidelines and Food Reformulation in The Netherlands

**DOI:** 10.3390/nu13124536

**Published:** 2021-12-17

**Authors:** Sovianne ter Borg, Elly Steenbergen, Ivon E. J. Milder, Elisabeth H. M. Temme

**Affiliations:** National Institute for Public Health and the Environment, 3721 MA Bilthoven, The Netherlands; sovianne.ter.borg@rivm.nl (S.t.B.); elly.steenbergen@rivm.nl (E.S.); ivon.milder@rivm.nl (I.E.J.M.)

**Keywords:** Nutri-Score, front-of-pack nutritional labelling, dietary guidelines, reformulation, The Netherlands

## Abstract

An unhealthy dietary pattern is an important risk factor for non-communicable diseases. Front-of-Pack nutritional labels such as Nutri-Score can be used to improve food choices. In addition, products can be improved through reformulation. The current study investigates to what extent Nutri-Score aligns with the Dutch Health Council dietary guidelines and whether it can be used as an incentive for reformulation. Nutri-Score calculations were based on the Dutch Branded Food database (2018). The potential shift in Nutri-Score was calculated with product improvement scenarios. The Nutri-Score classification is in line with these dietary guidelines: increase the consumption of fruit and vegetables, pulses, and unsalted nuts. It is, however, less in line with the recommendations to limit (dairy) drinks with added sugar, reduce the consumption of red meat and replace refined cereal products with whole-grain products. The scenario analyses indicated that a reduction in sodium, saturated fat or sugars resulted in a more favourable Nutri-Score in a large variety of food groups. However, the percentage of products with an improved Nutri-Score varied greatly between the different food groups. Alterations to the algorithm may strengthen Nutri-Score in order to help consumers with their food choices.

## 1. Introduction

An unhealthy dietary pattern is an important risk factor for non-communicable diseases [1]. To help consumers make informed, healthier food choices, the World Health Organization recommends product reformulation and the use of nutritional labelling [1,2,3]. Although food packages contain nutritional declarations at the back-of-pack there is a need for easier ways to inform the consumer about nutritional content. Front-of-Pack (FoP) labels are labels which are easy to understand and are clearly visible.

Nutri-Score is a FoP label developed in France by the National Epidemiology Research Institute [4] and was developed based on the existing evidence of the association between food consumption and chronic disease risk. In 2017, Nutri-Score was selected as the official national label in France [4]. Since then, Nutri-Score has been adopted by several European countries: Belgium, Spain, Germany, Switzerland, and Luxembourg [5]. In 2020, after considering different FoP labels [6], it was decided that Nutri-Score will be implemented in the Netherlands [7]. The use of Nutri-Score is voluntary in countries where it is being used. The Nutri-Score logo has two objectives: first, to guide consumers towards healthier food choices for packaged foods [4] and second to stimulate food reformulation [8].

Dietary guidelines and FoP labels such as Nutri-Score can be complementary and synergic measures, although based on different approaches and principles. Both are based on the existing evidence on associations between food consumption and risk and mortality from chronic diseases [9]. In the Netherlands, the evidence is summarized by the dietary guidelines of the Health Council [10]. Dietary guidelines identify food groups that are encouraged (e.g., fruits, vegetables, legumes etc.) and those that should be limited (e.g., red meat, salt), and recommend amounts for certain food groups (e.g., at least 200 g of vegetables and fruit daily) [10]. The Health Council dietary guidelines are subsequently translated into food based dietary guidelines, such as the Wheel of Five in the Netherlands. The Wheel of Five combines the food based dietary guidelines and daily recommended values into practical guidelines for consumers, to obtain healthy and complete food patterns, i.e., reducing the risk of disease and obtaining sufficient nutrients [11]. This translation covers a wider food consumption pattern and serves specific target groups such as children, men and (pregnant) women and older adults. In this study we take the guidelines from the Health Council as a starting point as these are closest to the evidence on the association between food consumption and chronic diseases. In addition, it enables comparison with international diet guidelines and studies.

Nutri-Score can be a helpful tool for consumers to identify healthier products and discriminate nutritional quality across and within packaged food groups with nutrient declarations [5]. Nutri-Score consists of five coloured categories (ranging from A: dark green to E: dark orange, Figure 1), which indicate the overall nutritional quality of the product within a food group [4,12,13]. The Nutri-Score is based on the United Kingdom Food Standards Agency Nutrient Profiling System. Points are allocated to the energy, sugar, saturated fat and sodium contents (‘negative point’), and for the fruit, vegetables, pulses and nuts, fibre, and protein content (‘positive points’). An algorithm is used to calculate an overall score which is subsequently categorized into five categories (A to E) representing the range of products with a higher nutritional quality to products with a lower nutritional quality. Slightly different algorithms are used for beverages, fats and oils, and cheeses. For additional details on the Nutri-Score calculation see [13].

In a study [5] in eight countries (i.e., Finland, France, Norway, Poland, Portugal, Slovakia, Sweden, Switzerland, but not including the Netherlands) the overall distribution of foods in the different Nutri-Score classes was compared with the food-based dietary guidelines from these countries and the general guidelines from the WHO. The study showed high discriminating ability (i.e., containing at least three Nutri-Score categories) for all food groups, with similar results in the different countries, and consistency with nutritional recommendations [5]. For instance, 91–96% of fruit and vegetable products were classified in the two healthiest Nutri-Score categories, while sugar (products) and butter/animal fats were mainly classified in the two less healthy categories of the Nutri-Score (88% in both groups respectively). For the food groups meat, meat products and milk, milk products and milk substitutes, however, differences were found in the Nutri-Score distribution between the countries. The authors state that this is possibly due to the large diversity of these food groups. Although previous studies in European countries have shown promising results, it is unclear to what extent Nutri-Score can discriminate packaged foods on their nutritional quality in line with the Dutch food-based dietary guidelines.

The other objective of Nutri-Score is to stimulate food product improvement [4]. The ability of a FoP logo to stimulate food reformulation was shown in previous research on the Dutch Choices logo [15]. For Nutri-Score, however, this is scarcely studied. The ratings of the FoP logo, based on the algorithm, may provide an incentive for the producer for, e.g., salt or sugar reduction or increased fibre content. For example, by lowering the salt content, a better Nutri-Score rating and colour may be reached.

Therefore, the aims of the current study are: (1) to investigate to what extent the Nutri-Score classification of foods and drinks aligns with the Dutch food-based dietary guidelines and (2) to perform theoretical scenario analyses as to whether Nutri-Score can be an incentive for reformulation of (un)favourable components.

## 2. Materials and Methods

### 2.1. Data Preparation for the Calculation of Nutri-Score

For the current analyses, food composition data from the Dutch Branded Food database [16,17] were used. This database contains the back-of-pack nutritional information and includes both private brands as well as supermarket brands. Data from 2018 were used. Food groups were defined according to the Dutch RIVM Reformulation Monitor 2018 [18]. Food groups not classified within the Reformulation Monitor (such as unprocessed meat) were created based on characteristics available within the Branded Food Composition Database. Nutri-Scores were calculated for foods with a mandatory nutrient declaration, as stated in the EU Regulation No 1169/2011 [19]. Nutri-Scores were also calculated for un/minimally processed fruit and vegetables and fresh meat products, for the evaluation of alignment with dietary guidelines. These product groups are at the moment not eligible for a Nutri-Score as they do not have a mandatory nutrient declaration. Other food groups for which the nutrient declaration is not mandatory (alcoholic drinks, coffee, tea, and eggs), alcohol-free beverages and nutritional supplements were excluded. A total of 52,357 records were included in the analyses.

Nutri-Score calculation guidelines [13] were followed. Walnut, rapeseed and olive oils in the food group ‘Oils and fats’ were assigned a Nutri-Score C. These oils can also be included as an ingredient in products. However, as the type of oil is not always listed on the nutrition label, these were not included in the Nutri-Score calculation. Dairy drinks were assumed to contain more than 80% milk and were considered as foods, following the calculation guidelines. Information on the content of fruit, vegetables, pulses and nuts (%) in foods was not available. Therefore, an estimation was made per food group, based on the ingredient lists (see Appendix A, Table A1).

Products with missing values (i.e., energy, sugar, saturated fat, sodium, protein, fibre, and fruit, vegetables, pulses and nuts content) were excluded, except for fresh fruit, fresh vegetables and bottled water: these products were assigned a Nutri-Score A. As declaration of fibre content is not mandatory, depending on the food groups there was a relatively high proportion of missing values. Therefore, for missing values the mean fibre content was imputed at food group level (see Appendix A, Table A2).

Outliers were removed at food group level by removing products with values below the first percentile and above the 99th percentile for nutrients needed for the calculation of the Nutri-Score. The Nutri-Score was computed based on the algorithm of Santé Publique France [14]. ‘Negative points’ (0–10) were allocated to the energy, sugar, saturated fat and sodium content, and ‘positive points’ (0–5) to the fruit, vegetables, pulses and nuts, fibre, and protein content. The overall score (range −15 to +40) was calculated by subtracting the ‘positive points’ from the ‘negative points’. If the total for fruits and vegetables was below five points, then the total points for protein was not subtracted from the ‘negative points’ for the calculation of the overall score. Depending on the score, one of the five Nutri-Score categories was assigned (A–E). ‘A’ represents foods with a higher nutritional quality; ‘E’ represents foods with a lower nutritional quality.

### 2.2. Evaluation of Aligment with Food-Based Dietary Guidelines

To determine to what extent the Nutri-Score is consistent with the dietary guidelines, the percentage of products per Nutri-Score category was calculated and compared to the Dutch food-based dietary guidelines 2015 [10,20]. These guidelines were derived from existing scientific evidence and include 15 specific guidelines on the consumption of foods and nutrients.

The dietary guidelines for food groups are to increase consumption (e.g., vegetables and fruits), to limit consumption (e.g., sugar-containing beverages), to moderate consumption (e.g., dairy) or to replace products (e.g., refined by whole grain cereal products or hard fats by oils) (see Table 1). The Nutri-Score category distribution was calculated for the food groups specified in the dietary guidelines. The dietary guidelines include the recommendation to limit the total salt intake (i.e., from foods and added by the consumer in the kitchen or at the table) to 6 g per day. The guidelines, however, do not specify which salt-rich foods should be limited. Bread, cheeses, meat preparations, savoury snacks and ready meals are mentioned as examples of products high in salt and are therefore included in the current analyses as foods to limit consumption of [10]. Other food groups, such as soups and sauces, contribute to the salt intake, but are not specifically mentioned in the guidelines, and are therefore not included in the analyses.

For the comparison between Nutri-Score and the guidelines, a similar method was used as described previously [21]. In case of good agreement with the dietary guidelines, it was expected that foods for which an increased consumption is recommended would mainly have a Nutri-Score A or B. Foods for which the consumption should be limited would mainly have a Nutri-Score D or E. The Nutri-Score and the dietary guidelines were considered to agree when >80% of the products in ‘increase consumption’ food groups scored an A or B and if >80% of the products in ‘limit consumption’ or ‘minimize consumption’ food groups scored a Nutri-Score D or E. The guidelines recommend the ‘replacement’ of refined cereal products with unrefined or whole-grain products to increase the fibre intake. To see whether fibre-rich products scored a more favourable Nutri-Score, the Nutri-Score categories in the product group ‘Cereals’ were analysed per quartile of fibre content. In addition, the fibre content was analysed per type of bread (e.g., whole grain) and Nutri-Score. The guidelines also recommend replacement of saturated by unsaturated fatty acids; therefore within the groups ‘Oils and fats’ the Nutri-Score distribution was calculated per quartile of ratio saturated fat/total fat. The Nutri-Score categories for ‘Meat preparations’, ‘Savoury snacks’ and ‘Ready meals’ were analyses per quartile of sodium content. The results of the quartile analyses were visually inspected.

### 2.3. Evaluation of Stimulation of Food Reformulation

To evaluate whether Nutri-Score could be an incentive for food reformulation, food groups with processed foods were selected based on their contribution to the total daily salt, saturated fat and sugar intake. The contribution to the total daily nutrient intake was based on the Dutch Food Consumption Survey 2012–2016 [22]. This survey includes a representative sample of the Dutch population aged 1–79 years (*n* = 4313). For fibre, a selection of food groups was chosen: cereal products and composite dishes. Food groups for which the contribution to the total daily nutrient (i.e., sodium, sugar, saturated fat) was assumed to be relevant (intake above 3%) and for which product reformulation on the specific nutrient is feasible were included. These include processed and composite products with added salt and/or sugar such as sauces, snacks, sweets, sugar-sweetened beverages and processed meats. The food groups are—to a large extent—the food groups of which consumption should be limited according to the Dutch food-based dietary guidelines.

The potential shift in Nutri-Score was calculated with theoretical product improvement scenarios. It was assessed whether a 1-point decrease in the Nutri-Score sub score could result in a more favourable Nutri-Score. As a reference, food compositions from the Dutch Branded Food database were used. In the sodium product improvement scenario sodium levels were modelled to decrease by 90 mg/100 g (1 point on the Nutri-Score sub score). In the saturated fat product improvement scenario, saturated fatty acid levels were decreased by 1 g/100 g (1 point on the Nutri-Score sub score). For the sugar product improvement scenario sugar levels were decreased −4.5 g for foods and −1.5 g for drinks per 100 g (−1 point on the Nutri-Score sub score). The product improvement scenario was also calculated for fibre content: scoring 1 point more (+0.9 g/100 g) on the Nutri-Score scale. All other components did not change. For all scenarios, the Nutri-Score for each food was recalculated and the number of foods per Nutri-Score were compared with the reference situation. The percentage of foods in each Nutri-Score category was calculated and compared with the reference situation for each score (A–E) as well as the total percentage of change.

## 3. Results

### 3.1. Evaluation of Alignment with Food-Based Dietary Guidelines

See Table 1 for an overview of the Nutri-Score distribution per food group. The dietary guidelines recommend an increase in the consumption of fruit and vegetables, legumes and unsalted nuts. The Nutri-Score is in line with these guidelines as the food groups fruit and vegetables and unsalted nuts and seeds were mainly classified as Nutri-Score A or B (93% and 90% respectively). All pulses were classified as Nutri-Score A.

A higher consumption of fish is recommended in the guidelines, preferably fatty fish. Nutri-Score is not in line with this recommendation as fish is classified as Nutri-Score A or B (61%), C (17%) and D (22%). Fish is a very heterogenic group, including both processed and unprocessed foods.

The guidelines recommend a more plant-based and a less animal-based dietary pattern. The consumption of red meat, particularly processed meat, should be limited. The Nutri-Score classification differs between the types of meat product. Unprocessed meats are mainly classified as Nutri-Score A or B (87% for red meat products, 97% for white meat products). Composed and single processed meat and meat preparations score mainly a Nutri-Score D or E (95% and 57%, respectively). The Nutri-Score classification of meat alternatives is not in line with the recommendation, as 61% score a Nutri-Score A or B (61%), and about one third (30%) received a C score. The C score is mostly because of a high salt content.

Dutch dietary guidelines recommend maintenance of dairy consumption (including milk and yoghurt). For dairy, Nutri-Score varies between the different types of dairy product (i.e., dairy drink, yoghurt or dessert). Dairy drinks and yoghurt, with no added sugar, are classified as a Nutri-Score A (100%). Dairy drinks and yoghurts with added sugar score Nutri-Score A or B (dairy drinks 94% and yoghurt 57%). Desserts score more frequently a Nutri-Score C or D and two percent (2%) of desserts with added sugar are classified as Nutri-Score E.

The consumption of sugar-containing beverages should be limited. Bottled water and soft drinks without added sugar mainly score a Nutri-Score A or B (100% and 91%, respectively). A large proportion of syrups, squash and cordial, fruit juice, fruit juice drink and soft drinks with added sugar score a C (27–57%).

The guidelines include the recommendation to limit salt intake. Cheeses, meat preparations, savoury snacks, bread and ready-to-eat meals are mentioned in the guidelines as examples of products high in (added) salt. When comparing the guidelines with the Nutri-Score classification, there is a mixed result. Cheeses are mainly classified as Nutri-Score D and E (99%). Bread is classified differently according to the type of bread: bread was classified as Nutri-Score A or B (95–100%); however, savoury bread is mainly classified as a Nutri-Score B and D (40% and 40%). Meat preparations and savoury snacks are often classified as a Nutri-Score D or E (57% and 66% respectively). Ready meals are often classified as Nutri-Score A or B (64%) and less frequently as Nutri-Score D or E (7%). When analysed per quartile of salt content, meat preparations, savoury snacks and ready meals with a lower salt content receive a more favourable Nutri-Score compared to products with a higher salt content (see Appendix A, Figure A1, Figure A2 and Figure A3).

To increase the fibre intake, the guidelines recommend replacement of refined products with unrefined or wholegrain products. Cereal products with the highest fibre content more frequently scored a Nutri-Score A or B (85%, see Appendix A, Figure A4), compared to products in the other quartiles. The Nutri-Score classification, however, does not discriminate between products high and low in fibre content: more than half of the cereal products were classified as a Nutri-Score A or B, independently of the fibre content (i.e., in all quartiles). Only a few cereal products were classified as Nutri-Score E (up to 3%). The Nutri-Score A and B classes contained wholegrain bread types, but also white bread types (see Appendix A, Figure A5).

Saturated fatty acids should be replaced with unsaturated fatty acids. None of the oils and fats was classified as a Nutri-Score A or B (see Appendix A, Figure A6). Oils and fats with a lower ratio of saturated fat to total fat content were classified as a Nutri-Score C or D (quartile 1 = 100%, quartile 2 = 95%), whereas oils and fats with a higher ratio were also classified as Nutri-Score E (quartile 3 = 36%, quartile 4 = 98%).

### 3.2. Ability to Stimulate Food Reformulation

Figure 2, Figure 3, Figure 4 and Figure 5 show the total percentage of foods that shifted to a more favourable Nutri-Score after reformulation of a nutrient with 1 point on the Nutri-Score sub score. For the complete overview of the Nutri-Score distribution per food group regarding the reference situation and product improvement scenario for reformulation, see Appendix A Table A3, Table A4, Table A5 and Table A6.

For sodium, the Nutri-Score (could be) improved in all food groups, except for pulses, when sodium content was decreased by one point (see Figure 2). The largest proportion of products that shifted to a more favourable Nutri-Score was found for soups (29%), composite dishes (24%) and nuts and seeds (20%). The potential improvement in Nutri-Scores was the lowest for meat, poultry and meat preparations (12%) and cheeses (7%). All processed pulses were classified as Nutri-Score A in the reference scenario, so no further improvement was observed after reformulation of sodium content. A one-point decrease in sodium content (90 mg/100 g) is relatively large for soups, as this is about a quarter (27%) of the median sodium content. For meat, poultry and meat preparations, a one-point decrease is smaller (11%) compared to the median content. A decrease of one point may therefore be more achievable for meats than for soups.

For saturated fat, the largest overall improvements in Nutri-Score by reformulation were found for spreads and cooking fats (19%), cereals (18%) and composite dishes (17%) (see Figure 3). For cheeses, potential improvements by lower saturated fat content were the smallest (0.2%). A one-point decrease in saturated fat content (i.e., −1 g per 100 g) is relatively large compared to the median content for cereals (63%) and composite dishes (67%), whereas this is smaller for spreads and cooking fats (3%). Decreasing the saturated fat content by one point may therefore be more achievable in spreads and cooking fats compared to cereals and composite dishes.

For sugar, the largest potential improvement in Nutri-Score was found for milk products (with added sugar) (31%), followed by sugary drinks and sweetened cereals (see Figure 4). The smallest overall improvement was found for sweetened baked goods and confectionery (6%). A one-point decrease in sugar content (i.e., −4.5 g per 100 g food and −1.5 g per 100 g for drinks) is relatively large compared to the median content in milk products (38%), whereas this is smaller in cereals (8%). Reducing the sugar content in cereals, by one point, may be achievable.

By reformulating the fibre content the Nutri-Score improved more for composite dishes (16%) than for cereals (12%) (see Figure 5). A one-point increase in fibre content (+0.9 g per 100 g) in composite dishes is relatively large (56%) compared to the median content. A one-point increase in fibre content in cereals is relatively small (23%) and therefore more feasible.

## 4. Discussion

In this paper we compared the Dutch food-based dietary guidelines with the Nutri-Score and explored whether the Nutri-Score can be (used as) an incentive for food reformulation.

### 4.1. Nutri-Score and the Food-Based Dietary Guidelines

Nutri-Score aims to guide consumers towards foods with a higher nutritional quality for packaged foods. Nutri-Score is based on a selection of specific nutrients (energy, sugars, saturated fat, sodium, fibre, protein) and other elements (percentage of fruit, vegetables, pulses and nuts), and does not include other aspects related to the healthiness of a food, such as portion-size, level of processing or additives.

Nutri-Score provides an overall evaluation as to whether a product had a higher or lower nutritional quality within a food group. For certain food groups, the Nutri-Score is in line with the food-based dietary guidelines. Fruit and vegetables, pulses, and unsalted nuts were mainly classified as Nutri-Score A or B (93%, 100% and 90%, respectively). Nutri-Score is less in line with the guidelines to limit the consumption of dairy drinks with added sugar, to reduce red meat and to replace refined grain products. These foods were also frequently classified as a Nutri-Score A, B or C rather than a Nutri-Score D or E.

Recently, a study using nutritional composition data from eight European countries (i.e., Finland, France, Norway, Poland, Portugal, Slovakia, Sweden, Switzerland) [5] and a similar study using German composition data [21] showed that Nutri-Score was mostly consistent with WHO and German dietary guidelines and that Nutri-Score discriminates foods, based on their nutritional quality, within a food group. They found similar results in the different countries. Our results are in line with these findings: fruits, vegetables and pulses were mostly classified in the Nutri-Score categories A/B, and cereals and nuts and seeds were mainly categorized as A to C. Oils and fats were mainly classified in the category D, with vegetable fats more frequently scoring a C and animal fats a Nutri-Score E. Although we did not analyse vegetable and animal fats separately, our analysis also indicates that products higher in saturated fat more often score a Nutri-Score E.

The dietary guidelines recommend following a more plant-based diet and consuming less animal-based products. The majority of plant-based foods such as fruits, vegetables and pulses were classified as expected. The majority of unprocessed meat (87–97%) were, however, classified as a Nutri-Score A–B, and are not in line with this recommendation. It must be noted that Nutri-Scores were calculated for un/minimally processed fruit and vegetables and unprocessed meat products, for the evaluation of alignment with dietary guidelines. These product groups are at the moment not eligible for a Nutri-Score as they do not have a mandatory nutrient declaration. To stimulate a more plant-based diet and limit the intake of (red) meat within the context of Nutri-Score, several aspects can be considered. First, fresh fruits and vegetables are not always packaged and with nutrient declaration and thus will not show a Nutri-Score. To guide the consumer to a more plant-based diet, showing Nutri-Score on all products, or on shelves, may be considered. Second, an option is to include the percentage of (red) meat in the product (like the percentage of saturated fat), in the Nutri-Score algorithm. As a result, meat will have a less favourable Nutri-Score. The ingredients list on the label might be used, as it is mandatory to report the percentage of meat in the product when greater than two percent [19]. Third, the Nutri-Score was assigned based on the nutrient declaration of unprepared meat as sold. Calculating the Nutri-Score based on prepared meat (including more salt and fat, based on a common cooking practice), may result in a less favourable Nutri-Score. Fourth, some of the meat alternatives scored moderately (i.e., Nutri-Score C) compared to certain types of meat, which does not stimulate consumers towards a more plant-based diet. Meat alternatives, however, can be reformulated to contain less salt and to shift to a more favourable Nutri-Score, as shown in our results.. Future research is needed on the feasibility of these suggested alterations. In addition, the effects on purchasing behaviour should be studied to see whether these alterations in Nutri-Score are able to stimulate a more plant-based diet. It should be warranted that alterations in the Nutri-Score algorithm do not result in a more favourable score for animal-based products and do not stimulate the consumption of such products.

Within the packaged meat and fish categories, Nutri-Score may discriminate between processed and raw/unprocessed foods [5]. Our results show that unprocessed meats scored more favourably than processed meats. A range of Nutri-Scores was found in the fish food group: 61% in A/B, 17% in C and 22% in D. This food group included both processed and unprocessed fish products.

Dréano-Trécant et al. [5] concluded that Nutri-Score can discriminate between refined and whole grain products in many of the countries studied. Our results, however, show that cereal foods often score an A or B, even though they have a low fibre content. To improve the ability to discriminate between whole and refined grain, the Nutri-Score algorithm might be adapted, increasing the number of points for fibre content, and so its contribution to the total sum score. To stimulate the consumption of fibre-rich wholegrain cereal products, the algorithm’s maximum fibre amount might be adapted. For fibre, the maximum amount for scoring ‘positive’ points is set at 3.5 g/100 g, and wholegrain bread and whole grain pasta contain 7 g/100 g. This may limit further product reformulation. Another option is to include whole grain as a positive component in the algorithm, instead of the fibre content, in order to stimulate the use of whole grains in products.

The Nutri-Score classification was partly in line with the recommendation to limit sugar-containing drinks. Water and soft drinks without added sugar had a favourable Nutri-Score, whereas drinks containing sugar frequently scored a Nutri-Score C. Dairy drinks with added sugar are not in line with the recommendation, and 94% were classified as A or B. Although it is recommended to maintain the consumption of dairy, this is not in line with minimizing the intake of sugar-containing drinks. Soft drinks with artificial sweeteners score favourably. Nutri-Score, however, promotes the consumption of water, rather than artificial sweetened beverages, by assigning a Nutri-Score A [13]. The Dutch health council diet guidelines did not include a recommendation concerning artificially sweetened drinks, but they advise alternatives: water, tea and unfiltered coffee without sugar [10,20].

The Dutch food-based dietary guidelines also include recommendations to increase the consumption of tea and replace coffee with filtered coffee. These were not included in the analyses, as these products do not contain a nutrition declaration on their packaging. The Dutch health council diet guidelines did not include recommendations on saturated fatty acids, trans fatty acids and added sugar in foods and are therefore not included in the analyses. Criteria for these nutrients are, however, included in the translated food based dietary guidelines of the Wheel of Five [11]. To help the consumer, it is important that Nutri-Score is in line with the Health Council dietary guidelines as well as with the Wheel of Five. Consistent communication and education are recommended during the implementation of the logo [5,23]. Posting health-oriented displays or slogans in sales locations may aid public awareness and understanding. The effectiveness of possible interventions must be further explored.

It must also be noted that many other aspects influence dietary choices, such as individual preferences and social, economic and environmental factors [24]. Previous online studies have indicated that Nutri-Score can help consumers to rank products in a similar product group, from healthy to less healthy options [8,25], and to choose more healthful products in an experimental setting [26]. Additional research is, however, needed on actual purchasing behaviour in a real-life setting [26], and the effect of Nutri-Score on the consumers’ eating pattern [25].

### 4.2. Nutri-Score’s Ability to Stimulate Food Reformulation

Nutri-Score may be an incentive for manufacturers to stimulate reformulation towards foods with a higher nutritional quality. In our scenario analyses, decreasing the sodium, saturated fat or sugar content by one point resulted in certain products shifting to a more favourable Nutri-Score. Nutri-Score may stimulate reformulation of, for instance, composite dishes towards a lower sodium and saturated fatty acids content, or of cereals with a lower saturated fat and sugar content and higher fibre content. The percentage of products with an improved Nutri-Score varied greatly, however, between the different product groups (ranging from about 0–30%). A one-point reduction in the Nutri-Score sub score does not necessarily result in a more favourable Nutri-Score (e.g., sodium reduction in single processed raw/cured meat). To shift the Nutri-Score in these groups, a larger reduction may be needed or multiple nutrients/components need to be reformulated. In addition, the reformulation scenarios showed that, for some food groups such as pulses and cheese, it might be more difficult to reformulate towards lower salt content since only one or two Nutri-Score categories were covered.

In the study with data from eight European countries [5], Nutri-Score was considered to be discriminating when at least three categories of the Nutri-Score were observed within a food (sub)group. For almost all food groups (except pulses), Nutri-Score was shown to be discriminating. We observed similar results based on the Dutch dataset: the classification of snacks and meals varied over four categories, which may help to choose the less salty and less sugary and fatty foods within these categories. The interpretation of sufficient discrimination differs, however. Van Tongeren et al. [27] concluded that Nutri-Score was not sufficiently discriminating, as not all five Nutri-Score categories were observed in a certain food subgroup [27]. They found a range of Nutri-Score categories for ready meals (A–E), soups (A–C, majority C), meal sauces (A–D, majority C) and cheese (A–E, majority D), comparable to our results.

It is unclear whether our calculated theoretical shift in Nutri-Score will also result in a more favourable Nutri-Score in a real-life setting of product improvement. We analysed a one-point difference in the scoring for a specific nutrient, assuming the remaining food composition remains constant. In reality, the Nutri-Score algorithm sums multiple (positive and negative) nutrients, and scoring better on one component (e.g., vegetables) may disguise another component (e.g., salt). In addition, reformulation of products, in order to score one point less on the Nutri-Score sub score, may not be practically feasible for all products. For soups and bread, for instance, a one-point reduction in sodium (−90 mg per 100 g) is relatively large compared to the median salt content of these products. Reducing the size of a one-point step on the Nutri-Score sub scale may increase feasibility and stimulate reformulation. Even though a one-point reduction in nutrient content will be smaller, the overall impact on the intake may be considerable, as these products are regularly consumed. It will be worthwhile to monitor changes in food composition before and after the introduction of Nutri-Score in the Netherlands to see the effect on consumers’ food choices as well as producers’ food product improvement.

### 4.3. The Nutri-Score Algorithm

Nutri-Score shows an overall assessment of multiple nutrients/components, which makes it easy to interpret for the consumer. Its design, summing ‘positive’ and ‘negative’ components, also has certain implications. Certain nutrients may compensate for others. Dairy drinks with added sugar, for instance, are generally low in salt and saturated fat, but high in protein, resulting in less than 11 ‘negative’ points. Due to the protein content, they score relatively high on the ‘positive’ points. As a consequence, when subtracting the ‘positive’ from the ‘negative points’, the final score is relatively low, resulting in a favourable Nutri-Score. The protein content therefore may compensate for the sugar content. In addition, some foods may be classified as having a lower nutritional quality, as they do not contain both the ‘positive’ and ‘negative’ components of the algorithm. In the present study 63% of oils and fats scored a D or E. These foods do not shift towards a more favourable Nutri-Score as they are high in ‘negative’ points (energy, saturated fat) and low in ‘positive’ points (e.g., fruits, vegetables). The algorithm, however, partially takes this into account by including specific cut-off points for fats [4]. For certain foods, the summing of ‘positive’ and ‘negative’ components has a positive effect on the Nutri-Score. As vegetables score low on the ‘negative’ points (salt, sugar, saturated fat) and high on the ‘positive’ points (vegetable component) they mainly score an A. For composite dishes, the summing in the algorithm provides an opportunity to improve food composition, as they can be reformulated in both the ‘negative’ (decrease saturated fat, sugar and salt) and ‘positive’ (increase vegetable, fibre) components.

### 4.4. Strengths and Limitations of the Current Study

For the current analyses, the Dutch Branded Food database was used. This database contains a large amount of nutritional information for products available in Dutch supermarkets. It is estimated that the data in the Branded Food database represent about 75% of the products available in supermarkets. The results of the analyses therefore provide a robust assessment of products on the Dutch market. It must however be noted that the database contains data which are voluntarily supplied by the manufacturers and retailers. Although quality checks are performed, the completeness of the data is dependent on the data provided by the manufacturers. In addition, the database does not reflect the market share of the products. It is unclear which method the manufacturers used to assess the fibre content in their product, and consequently in the Branded Food database. For the current analyses the Association of Official Analytical Chemists (AOAC) method was used, as advised, for calculating the Nutri-Score [13]. We were unable to distinguish certain product groups, such as fatty fish and refined and whole-grain breakfast cereals. In addition, rice and pasta were not included. Adding these food groups would strengthen the interpretation of Nutri-Score and the guidelines. Unpackaged products, such as vegetables and fruits, are not included in the Branded Food database. These products are therefore underestimated in the current analyses. Still, a large amount of packaged vegetables and fruits were included in the analyses, providing valuable insight into the Nutri-Score classification of these healthier products.

It must be noted that we did not study the effect on consumer behaviour, and the effects on actual purchasing behaviour.

## 5. Conclusions

The Nutri-Score classification is in line with the Dutch food-based dietary guidelines: increase the consumption of fruit and vegetables, pulses, and unsalted nuts. It is however less in line with the recommendations to limit (dairy) drinks with added sugar, to reduce the consumption of red meat and to replace refined cereal products with whole-grain products. In our scenario analyses, decreasing the sodium, saturated fat or sugar content by one Nutri-Score point resulted in certain products (e.g., composite dishes, cereals, milk products) shifting towards a more favourable Nutri-Score (ranging from about 0–30%). Nutri-Score may therefore be an incentive for reformulation of certain foods. Alterations to the algorithm may strengthen Nutri-Score in order to help consumers with their food choices.

## Figures and Tables

**Figure 1 nutrients-13-04536-f001:**
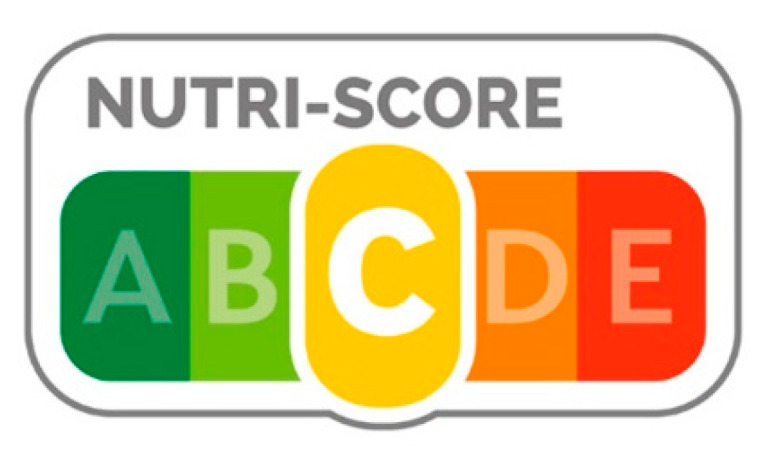
The Nutri-Score logo/Santé publique France 2017. This FoP label indicates the nutritional quality of a food product based on five coloured categories (A–E) [14].

**Figure 2 nutrients-13-04536-f002:**
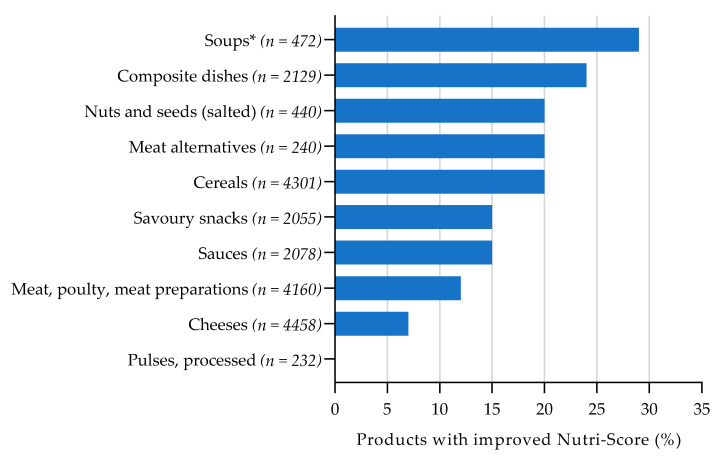
Estimated effect of sodium reformulation on Nutri-Score, assuming the levels of the other food components remain constant. Bars represent the proportion of products with an improved Nutri-Score when scoring 1 point less (−90 mg/100 g) on the Nutri-Score sub score for sodium. * Soups including broth and stocks.

**Figure 3 nutrients-13-04536-f003:**
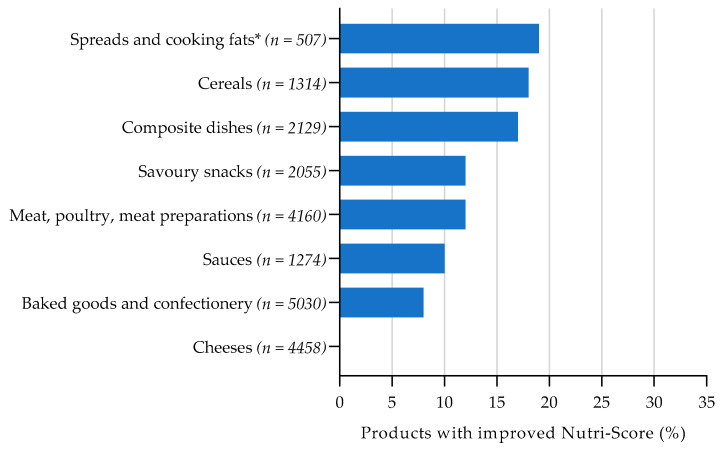
Estimated effect of saturated fat reformulation on Nutri-Score, assuming the levels of the other food components remain constant. Bars represent the proportion of products with an improved Nutri-Score when scoring 1 point less (−1 g/100 g) on the Nutri-Score sub score for saturated fat. * 1 Nutri-Score point/median ratio % saturated fat content.

**Figure 4 nutrients-13-04536-f004:**
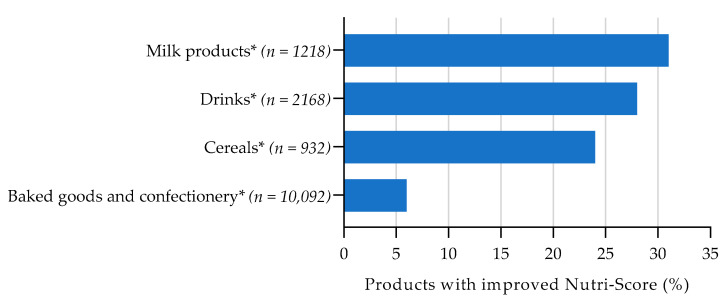
Estimated effect of sugars reformulation on Nutri-Score, assuming the levels of the other food components remain constant. Bars represent the proportion of sweetened products with an improved Nutri-Score when scoring 1 point less (−1.5 g/100 g in drinks and −4.5 g/100 g in food and dairy drinks) on the Nutri-Score sub score for sugars. * With added sugar.

**Figure 5 nutrients-13-04536-f005:**
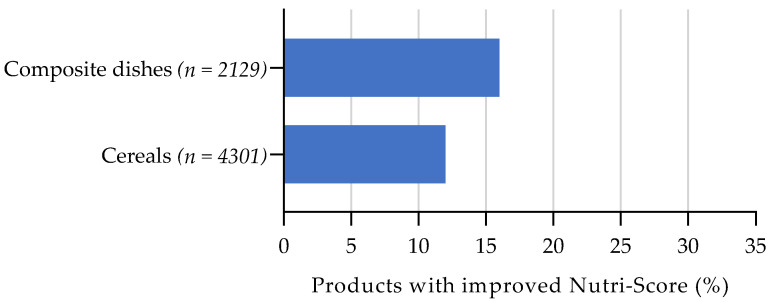
Estimated effect of fibre reformulation on Nutri-Score, assuming the levels of the other food components remain constant. Bars represent the proportion of products with an improved Nutri-Score when scoring 1 point more (+0.9 g/100 g) on the Nutri-Score sub score for fibre.

**Table 1 nutrients-13-04536-t001:** Foods classified according to the Nutri-Score and comparison with the Dutch food-based dietary guidelines [10].

Food Group ^1^	Food-Based Dietary Guideline ^2^	N	Nutri-Score (%)				
			A	B	C	D	E
General guideline	Guidelines: follow a dietary pattern that involves eating more plant-based food and less animal-based food. Limit salt intake to 6 g daily. Comparison analysis: eat less meat by replacing with meat-alternatives. Limit the consumption of salt-rich products (such as bread, cheeses, meat preparations, savoury snacks and ready meals)						
Fruit and vegetables ^3^	Guideline: eat at least 200 grams of vegetables and at least 200 grams of fruit daily. Comparison analysis: higher consumption recommended	7225	89	4	7	0	0
Protein-rich products							
Meat, poultry, meat preparations ^3^	Guideline: limit the consumption of red meat, particularly processed meat. Limit salt intake to 6 g daily Comparison analysis: limit the consumption of red meat, particularly processed meat. Limit the consumption of salt-rich products	5557	20	10	8	33	30
Meat preparations		1374	6	16	21	47	10
Composed and single processed meat ^4^		2786	0	1	4	40	55
Meat unprocessed, red meat ^3^		546	64	23	3	10	0
Meat unprocessed, white meat ^3^		851	78	19	2	0	0
Meat alternatives		240	50	11	30	9	0
Milk products	Guideline: take a few portions of dairy produce dialy, including milk and yoghurt Comparison analysis: maintain the current consumption, including milk and yoghurt	1392	17	36	32	14	1
Dairy drinks, with added sugar	Guideline: minimize the consumption of sugar-containing drinks Comparison analysis: see guideline	320	11	83	6	0	0
Dairy drinks, no added sugar		98	96	4	0	0	0
Yoghurt and quark, with added sugar		440	12	45	42	0	0
Yoghurt and quark, no added sugar		45	91	9	0	0	0
Desserts, with added sugar		458	0	5	54	39	2
Desserts, no added sugar		31	32	26	0	42	0
Cheeses	Guidelines: limit salt intake to 6 g daily Comparison analysis: limit the consumption of salt-rich products	4458	0	0	1	87	12
Pulses ^3^	Guideline: eat legumes weekly Comparison analysis: higher consumption of legumes recommended	301	100	0	0	0	0
Nuts and seeds	Guideline: eat at least 15 g of unsalted nuts daily Comparison analysis: higher consumption unsalted nuts recommended	1081	27	42	30	1	0
Nuts and seeds (unsalted)		641	39	51	10	0	0
Nuts and seeds (salted)		440	10	30	59	2	0
Carbohydrate- and fibre-rich products							
Cereal products	Guidelines: replace refined cereal products with whole-grain products. Eat at least 90 grams of brown bread, wholebread or other whole-grain products daily. Limit salt intake to 6 g daily Comparison analysis: replace refined cereal products with whole-grain products. Limit the consumption of salt-rich products	4301	37	28	19	15	2
Bread wheat-wholegrain		287	99	1	0	0	0
Bread miscellaneous		1861	47	48	5	0	0
Bread luxury, plain and sweet		467	1	4	46	43	6
Bread luxury, savoury		169	4	40	13	40	4
Bread alternatives (toast, crackers, etc.)		780	20	12	29	33	5
Bases (wraps, pizzabases, etc.)		213	7	25	33	23	11
Breakfast cereals		465	41	9	40	10	0
Oils and fats	Guideline: replace butter, hard margarines and cookong fats by soft margarines, liquid cooking fats and vegetable oils Comparison analysis: replace hard fats with liquid cooking fats and vegetable oils	801	0	0	38	31	32
Oils		294	0	0	69	27	3
Spreads, baking and cooking fats		507	0	0	19	33	48
Fish ^3^	Guideline: eat one serving of fish, preferably fatty oily fish, weekly Comparison analysis: higher consumption recommended, preferably oily fish	1593	25	36	17	22	0
Drinks	Guideline: minimize the consumption of sugar-containing drinks Comparison analysis: see guideline	3822	9	16	36	36	3
Bottled water		297	100	0	0	0	0
Syrups Squash and Cordial		314	0	10	29	27	34
Fruit Juice		698	0	0	27	73	0
Fruit Juice drink		429	0	3	55	42	0
Soft drinks, with added sugar ^5^		1425	0	0	57	42	1
Soft drinks, no added sugar ^5^		621	0	91	9	0	0
Vegetable juice		38	84	13	3	0	0
Savoury snacks	Guidelines: limit salt intake to 6 g daily Comparison analysis: limit the consumption of salt-rich products	2055	0	5	29	48	18
Ready meals ^6^	Guidelines: limit salt intake to 6 g daily Comparison analysis: limit the consumption of salt-rich products	2129	20	44	29	7	0

^1^ Tea, coffee, alcoholic beverages and egg are not included as these products do not contain a nutrition declaration; ^2^ dietary guidelines which are not included: increase the consumption of tea, replace coffee with filtered coffee, do not drink alcohol or no more than one glass daily; ^3^ including processed and unprocessed (not eligible for Nutri-Score) products; ^4^ including prepared and raw/cured processed meats; ^5^ including sport drinks and energy drinks; ^6^ including pizzas.

## Data Availability

The data presented in this study are available on request from the corresponding author. The crude data are not publicly available due to legal issues.

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
