# Peer review of "Evaluation of Nutri-Score in Relation to Dietary Guidelines and Food Reformulation in The Netherlands"

_nutrients, 2021, doi:10.3390/nu13124536_

Round 1

Reviewer 1 Report

The paper analyses how Nutri-Score is reflected in the packaged foods in the Netherlands, test the Nutri score against the Dutch dietary recommendations and examines the potential impact of reformulation on the Nutri Score. The authors present a critical look at the score, offer suggestions for improvement and grasp the limitations of the reformulation analyses, which is a purely theoretical discussion at this stage.

I find that the Nutri Score is limited to nutrients only and does not capture the total health value of the products. In order to discriminate healthy from unhealthy foods, the degree of processing must also be taken into account (food additives and technological processes), as extensive current literature shows. Except for meat, the Nutri Score does not include this dimension, so the discussion must be limited to the nutrients health value of the foods and not the total healthiness of the foods. This should be addressed in the text.

The overall impression is that the authors recommend some adaption and changes in the Nutri-Score to create a more significant reflection of the recommendations, and I could not agree more. The score of A or B to unprocessed red and white meat contradicts the pant-based recommendation. It should be emphasized that corrections should be made not to encourage increased consumption of such products. This is also should be critically evaluated.

Specific comments:

  1. What is included in meat alternatives? There are alternative which is ultra-processed) such as beyond meat, impossible meat, and alike), and other alternatives like tofu, for example, or insect powder, which are less processed. The processing level is crucial to estimate the health effects of these products over time.
  2. Page 5 Row 230: I understand that artificially sweetened beverages get a high score. There is evidence of adverse health effects of artificial sweeteners (although not as bad as sugar, still alarming); I think it is important to address this in the discussion as another Nutri-score limitation. It is possible that the algorithm also needs to be improved here, and it may be addressed in the discussion.
  3. Page 5 row 241: are ready meals equal to "TV meals"/ frozen meals?  If so, it is encouraged by the Nutri Score, probably because only salt is taken into account. This is another example of the limitations of the algorithm of the Nutri score. These meals are not recommended because they are usually ultra-possessed, high in fat content, and have less nutritional value.
  4. In the discussion, page 11 rows 330-335: you may consider adding a comment to this conclusion. While Nutri Score may discriminate foods with healthy nutrients from foods with less healthy nutrients or without healthy nutrients, it is limited in its ability to discriminate between healthy and unhealthy foods in a comprehensive approach. Therefore, as long as it does not include reference to non-nutrients, such as food additives and processing level (except in meat), which affect its healthiness, it is not correct to conclude about discriminating between healthy and non-healthy foods.

Proofreading corrections:

  1. Page 1 row 19: add "to" before "make healthier"
  2. Page 2 row 62- "points"?
  3. Page 6 row 248 - A "or" B
  4. Page 14 row 472 – seems like "of" is missing after "the interpretation

Author Response

Dear reviewer,

Thank you for your interest and for taking the time to review our manuscript.

Based on your suggestions we made several changes to the manuscript. Please find our response below in blue. In the manuscript we used track changes, to indicate our alterations.

Kind regards.

The paper analyses how Nutri-Score is reflected in the packaged foods in the Netherlands, test the Nutri score against the Dutch dietary recommendations and examines the potential impact of reformulation on the Nutri Score. The authors present a critical look at the score, offer suggestions for improvement and grasp the limitations of the reformulation analyses, which is a purely theoretical discussion at this stage.

I find that the Nutri Score is limited to nutrients only and does not capture the total health value of the products. In order to discriminate healthy from unhealthy foods, the degree of processing must also be taken into account (food additives and technological processes), as extensive current literature shows. Except for meat, the Nutri Score does not include this dimension, so the discussion must be limited to the nutrients health value of the foods and not the total healthiness of the foods. This should be addressed in the text.

We have included a comment in the discussion, that Nutri-Score does not include all aspects of a healthy food such as the degree of processing or additives (lines 329-333). We also changed the wording throughout the manuscript, replacing healthy/non-healthy with higher or lower nutritional quality of foods.

The overall impression is that the authors recommend some adaption and changes in the Nutri-Score to create a more significant reflection of the recommendations, and I could not agree more. The score of A or B to unprocessed red and white meat contradicts the pant-based recommendation. It should be emphasized that corrections should be made not to encourage increased consumption of such products. This is also should be critically evaluated.

In the discussion, we make several suggestions on how to improve Nutri-Score, to stimulate a more plant-based diet (lines 354-376). We have added the comments that additional research is needed on whether these adaptations are effective, and that alterations should not result in encouraging the consumption of animal-derived products (lines 379-381).

Specific comments:

1. What is included in meat alternatives? There are alternative which is ultra-processed) such as beyond meat, impossible meat, and alike), and other alternatives like tofu, for example, or insect powder, which are less processed. The processing level is crucial to estimate the health effects of these products over time.

The meat alternatives included both ultra-processed products and less processed products. We did not categorize the products based on the degree of processing, as this was outside of the scope if our manuscript.

2. Page 5 Row 230: I understand that artificially sweetened beverages get a high score. There is evidence of adverse health effects of artificial sweeteners (although not as bad as sugar, still alarming); I think it is important to address this in the discussion as another Nutri-score limitation. It is possible that the algorithm also needs to be improved here, and it may be addressed in the discussion.

We have included a section on artificial sweeteners (lines 406-411).

3. Page 5 row 241: are ready meals equal to "TV meals"/ frozen meals?  If so, it is encouraged by the Nutri Score, probably because only salt is taken into account. This is another example of the limitations of the algorithm of the Nutri score. These meals are not recommended because they are usually ultra-possessed, high in fat content, and have less nutritional value.

The Nutri-Score algorithm includes several nutrient/food component for this type of products: salt, sugar and saturated fat, but also the amount of vegetables, protein and fibre (see lines 494-497). Nutri-Score may therefore stimulate reformulation to a healthier product. However, as indicated above, Nutri-Score does not take the degree of processing into account.

4. In the discussion, page 11 rows 330-335: you may consider adding a comment to this conclusion. While Nutri Score may discriminate foods with healthy nutrients from foods with less healthy nutrients or without healthy nutrients, it is limited in its ability to discriminate between healthy and unhealthy foods in a comprehensive approach. Therefore, as long as it does not include reference to non-nutrients, such as food additives and processing level (except in meat), which affect its healthiness, it is not correct to conclude about discriminating between healthy and non-healthy foods.

We have included a comment in the discussion, that Nutri-Score is based on a selection of certain nutrients and does not include other aspects related to the healthiness of a food (lines 330-333). We also changed the wording healthy/non-healthy with higher and lower nutritional quality, to emphasize that Nutri-Score doesn’t reflect the overall healthiness of a food.

Proofreading corrections:

  1. Page 1 row 19: add "to" before "make healthier"
  2. Page 2 row 62- "points"?
  3. Page 6 row 248 - A "or" B
  4. Page 14 row 472 – seems like "of" is missing after "the interpretation

Thank you for these remarks. I’ve corrected the manuscript.

Reviewer 2 Report

Comments for nutrients-1480301 attached.

Author Response

Dear reviewer,

Thank you for your interest and for taking the time to review our manuscript.

Based on your suggestions we made several changes to the manuscript. Please find our response below in orange. In the manuscript we used track changes, to indicate our alterations.

Kind regards.

Reviewer 3 Report

The manuscript of ter Borg et al.  “Evaluation of Nutri-Score in relation to dietary guidelines and food reformulation in the Netherlands” addresses the Nutri-Score, a front of pack label that gained high importance in Europe. The authors investigated to what extent Nutri-Score aligns with the Dutch Health Council dietary guidelines and whether it can be used as an incentive for reformulation.

I have major concerns with the approach described in the manuscript. The study uses the Nutri-Score beyond its scope and intention. The intention of the Nutri-Score is “… not to separate 'good' foods from 'bad' foods, but rather to use the 5 classes to distinguish foods that are healthier from those that are less healthy….” https://www.santepubliquefrance.fr/media/files/02-determinants-de-sante/nutrition-et-activite-physique/nutri-score/qr-scientifique-technique-en. This in particular within one food category as the approach is based on 100g and does not consider the serving size, as would dietary guidelines. Resulting, the results, discussion and conclusion are disputable and the recommendations how to change the algorithm of the Nutri-Score are not well enough substantiated. Even though a similar approach was ones published (de Edelenyi et al. 2019), the conclusions were phrased differently, and the focus of the paper was the discriminating power of the Nutri-Score. The Nutri-Score should go hand in hand with dietary guidelines to complement them and enable suitable food choices but it cannot replace food group recommendations.  

A second concern is the very theoretical approach to study the reformulation potential which might not capture all aspects that the food industry would take into consideration for the decision of reformulation. A more practical approach based for example on available product compositions in the market seems more promising.

A strength of the paper is a good data base with the Dutch Branded Food database. Analysing the discriminating power of the Nutri-Score in a local setting (the Netherlands) is a relevant analysis and this analysis could be worth publishing if the aspects above and in the detailed comments are revised in major form. 

Further comments - Line:

35-36: several of these countries have adopted on a voluntary basis – this should be mentioned

36 – 37: add the year of decision for the Netherlands. This is relevant for the evaluation of reformulation potential. Add whether the display is mandatory.

47: dietary guidelines also include recommended amounts of food groups (e.g. Eat at least 200 grams of vegetables and at least 200 grams of fruit daily.)

84 – 87: from the previous sentences, the impression arises whether Nutri-Score can compare between food categories. However, Nutri-Score should mainly be used to compare within a category to facilitate consumer choices.

90: The coloured categories of the FoP logo – would replace with “the ratings of the FoP logo”

96 -97: would it not have been possible to study the change of product composition after introduction? Why a theoretical scenario? It is mentioned later in the text that a practical study would be useful – explain for the reader why a practical study is not possible (yet).

130 following: the explanation is incorrect.  “unfavourable” components as calories, sugars, sodium and saturated fatty acids get negative points and favourable ones positive points. Positive points are subtracted from negative points. See https://www.santepubliquefrance.fr/media/files/02-determinants-de-sante/nutrition-et-activite-physique/nutri-score/qr-scientifique-technique-en. In de Edelenyi et al. 2019 while the inverse explanation (positive and negative points) to the original French guide was used, a sum instead of subtraction was mentioned which turns the result correct. Change throughout the paper (e.g. 441 – 456)

155-157: not to include soups because they are not mentioned in the guidelines although they contribute to salt consumption is suprising

166 following:  whole grain has not only fiber as active component but also minerals and phytonutrients. To reduce whole grain to fiber content does not capture the whole potential. It is a draw-back of the Nutri-Score not to include also whole grain content – and is difficult to analyse and estimate but if a proposition of changes to the algorithms are made, that could be one. It should be avoided that the food industry adds certain fibers that ight not have the desired health benefit to improve the score.

191 – 202: was is verified that the modified product compositions are feasible from a technological and sensorial point of view i.e. are there products on the market that have these compositions? Otherwise the approach seems very theoretical.

219-22: why is the conclusion that this not in alignment? Meat alternatives are highly processed products and the sodium content is an important issue.

250 products

246 – 254: breakfast cereal products rich in fiber have often also higher sugar content to “mask” the fiber content. The nutrients cannot be considered isolated i.e. cereals only reduced to fiber content.

296: wrong wording: use increase (in fibre..)  instead of decrease

323 – 324: the distinction is between healthier and less healthy foods. This is not meant to go across categories.

346 – 347: were these suggestions tested on a more substantiated scale to see the effects on the ratings? Proposed changes to the algorithm should be better substantiated.

348-349: as it is with other ingredients it will be complicated to calculate the content of meat in a food product

352 – 353: meat alternatives are not necessarily healthy products if they are highly processed, contain a high content of salt, potentially low quality protein and mineral and vitamin content that is below that of meat

357-358: showing the Nutri-Score on unpacked foods is a big challenge. Again, the Nutri-Score should go hand in hand with dietary guidelines to complement them.  

371 – positive points (not negative)

369 – 372: why not suggest to add specifically the whole grain content as a beneficial component – see above

422 – 435: these are important comments that limit the value of the analyses. Analyses of product compositions of the nutritionally best products on the markets would have been more meaningful.

442: positive points

457 – 478: a certain limitation of the Nutri-Score is the reference base of 100 g that makes a product comparable within a food category but not as part of a diet. This is important for the consideration of oils for example. The limitations that were mentioned for the theoretical reformulation approaches should be mentioned here, too. In addition, for food companies it might not be the same to reformulate a product from B to reach an A but rather to avoid an E or D. It can be expected that products with different Nutri-Score ratings are not reformulated in equal amounts. Also, consumer insight is very important for these decisions. All these aspects would be expected here.

Author Response

Dear reviewer,

Thank you for your interest and for taking the time to review our manuscript.

Based on your suggestions we made several changes to the manuscript. Please find our response below in blue. In the manuscript we used track changes, to indicate our alterations.

Kind regards.

Comments and Suggestions for Authors

The manuscript of ter Borg et al.  “Evaluation of Nutri-Score in relation to dietary guidelines and food reformulation in the Netherlands” addresses the Nutri-Score, a front of pack label that gained high importance in Europe. The authors investigated to what extent Nutri-Score aligns with the Dutch Health Council dietary guidelines and whether it can be used as an incentive for reformulation.

I have major concerns with the approach described in the manuscript. The study uses the Nutri-Score beyond its scope and intention. The intention of the Nutri-Score is “… not to separate 'good' foods from 'bad' foods, but rather to use the 5 classes to distinguish foods that are healthier from those that are less healthy….” https://www.santepubliquefrance.fr/media/files/02-determinants-de-sante/nutrition-et-activite-physique/nutri-score/qr-scientifique-technique-en. This in particular within one food category as the approach is based on 100g and does not consider the serving size, as would dietary guidelines. Resulting, the results, discussion and conclusion are disputable and the recommendations how to change the algorithm of the Nutri-Score are not well enough substantiated. Even though a similar approach was ones published (de Edelenyi et al. 2019), the conclusions were phrased differently, and the focus of the paper was the discriminating power of the Nutri-Score. The Nutri-Score should go hand in hand with dietary guidelines to complement them and enable suitable food choices but it cannot replace food group recommendations.

We agree that the Nutri-Score is not designed to classify foods in either healthy or unhealthy, but to indicate a range. To emphasize this we changed the wording healthy in nutritional quality throughout the manuscript. The introduction includes a remark on Nutri-Score and dietary guidelines, that they are complementary measures (see lines 42-43).

A second concern is the very theoretical approach to study the reformulation potential which might not capture all aspects that the food industry would take into consideration for the decision of reformulation. A more practical approach based for example on available product compositions in the market seems more promising.

The manuscript is indeed based on a theoretical approach. De data itself however represent actual food compositions of packaged products available in Dutch supermarkets (lines 103-106 and 499-503). We discuss the theoretical approach and the unknown effects in a real-life setting (lines 461-475). We suggest to monitor the food composition changes before and after the introduction of Nutri-Score in the Netherlands, to see the effect of Nutri-Score on food reformulation efforts.

A strength of the paper is a good data base with the Dutch Branded Food database. Analysing the discriminating power of the Nutri-Score in a local setting (the Netherlands) is a relevant analysis and this analysis could be worth publishing if the aspects above and in the detailed comments are revised in major form. 

Further comments - Line:

35-36: several of these countries have adopted on a voluntary basis – this should be mentioned

36 – 37: add the year of decision for the Netherlands. This is relevant for the evaluation of reformulation potential. Add whether the display is mandatory.

We have added these comments in the manuscript (lines 36-39)

47: dietary guidelines also include recommended amounts of food groups (e.g. Eat at least 200 grams of vegetables and at least 200 grams of fruit daily.)

This comment was added in lines 46-49.

84 – 87: from the previous sentences, the impression arises whether Nutri-Score can compare between food categories. However, Nutri-Score should mainly be used to compare within a category to facilitate consumer choices.

The sentence now includes ‘within packaged food groups’ (lines 87-90).

90: The coloured categories of the FoP logo – would replace with “the ratings of the FoP logo”

We agree that it is not the colour, but the rating itself. We have changed the wording (lines 93-95). 

96 -97: would it not have been possible to study the change of product composition after introduction? Why a theoretical scenario? It is mentioned later in the text that a practical study would be useful – explain for the reader why a practical study is not possible (yet).

The theoretical scenario was the first step, to investigate how the Nutri-Score algorithm rates the different food groups and whether a shift in the algorithm score would result in a shift in the final rating of Nutri-Score. Nutri-Score is currently not yet implemented in the Netherlands. When it is implemented, we will perform an additional study to see the impact of Nutri-Score on the product composition. We briefly mention this in the discussion (lines 473-475).

130 following: the explanation is incorrect.  “unfavourable” components as calories, sugars, sodium and saturated fatty acids get negative points and favourable ones positive points. Positive points are subtracted from negative points. See https://www.santepubliquefrance.fr/media/files/02-determinants-de-sante/nutrition-et-activite-physique/nutri-score/qr-scientifique-technique-en. In de Edelenyi et al. 2019 while the inverse explanation (positive and negative points) to the original French guide was used, a sum instead of subtraction was mentioned which turns the result correct. Change throughout the paper (e.g. 441 – 456)

Thank you for this comment. In our manuscript we followed the wording as used by Julia et al. 2016 Plos one, however the interpretation is difficult as the wording ‘positive’ does not reflect the nutrient quality (sugars, sodiums etc.). We have changed the wording throughout the manuscript.

155-157: not to include soups because they are not mentioned in the guidelines although they contribute to salt consumption is suprising

They are not specifically mentioned in the guidelines. For your interest: soups are included in the reformulation effort by the Dutch food industry.

166 following:  whole grain has not only fiber as active component but also minerals and phytonutrients. To reduce whole grain to fiber content does not capture the whole potential. It is a draw-back of the Nutri-Score not to include also whole grain content – and is difficult to analyse and estimate but if a proposition of changes to the algorithms are made, that could be one. It should be avoided that the food industry adds certain fibers that ight not have the desired health benefit to improve the score.

This suggestion was added to the discussion (lines 399-400).

191 – 202: was is verified that the modified product compositions are feasible from a technological and sensorial point of view i.e. are there products on the market that have these compositions? Otherwise the approach seems very theoretical.

The findings are based on actual food compositions available in Dutch supermarkets, see above. We have included the feasibility in our discussion (lines 461-475). Soups are mentioned as an example, as a 1 Nutri-Score point reduction in the salt content (90mg/100g) is relatively large compared to the median salt content of soups.

219-22: why is the conclusion that this not in alignment? Meat alternatives are highly processed products and the sodium content is an important issue.

250 products

To be in line with the recommendation to consume more plant-based products, the classification in Nutri-Score A+B needed to exceed 80% (our cut-off used). As the score was 61% we concluded that it is not in line with the recommendation. In the discussion we mention the high salt-content, and the need for product reformulation in this food group (lines 373-376). 

246 – 254: breakfast cereal products rich in fiber have often also higher sugar content to “mask” the fiber content. The nutrients cannot be considered isolated i.e. cereals only reduced to fiber content.

Indeed, certain nutrient may mask other nutrients, which we comment on in the discussion (lines 479-485). As an example we included dairy drink with added sugar.

The cereal product which we refer to in the results section include not only breakfast cereal products, but also other cereal products such as bread.

296: wrong wording: use increase (in fibre..)  instead of decrease

We have changed the wording (lines 301-303).

323 – 324: the distinction is between healthier and less healthy foods. This is not meant to go across categories.

We have changed the wording (lines 334-335).

346 – 347: were these suggestions tested on a more substantiated scale to see the effects on the ratings? Proposed changes to the algorithm should be better substantiated.

We didn’t test these suggested alterations to the algorithm. We included a comment that additional research is needed to see the effects of these alterations (lines 379-382).

348-349: as it is with other ingredients it will be complicated to calculate the content of meat in a food product

Yes, we agree. We have added a comment that we need to study the feasibility (lines 379-380).

352 – 353: meat alternatives are not necessarily healthy products if they are highly processed, contain a high content of salt, potentially low quality protein and mineral and vitamin content that is below that of meat

This is in line with our findings: we have noticed that meat alternatives frequently scored a Nutri-Score C, due to the high salt content (lines 373-376). We have added a comment that processing is not part of the Nutri-Score (lines 330-333).

357-358: showing the Nutri-Score on unpacked foods is a big challenge. Again, the Nutri-Score should go hand in hand with dietary guidelines to complement them.

Yes, we agree that it should go hand in hand with the guidelines, and that Nutri-Score is not designed to replace the guidelines (lines 42-43). There is however a need for clear communication towards the consumer at the moment of purchasing (lines 418-423), which includes unprocessed foods.

371 – positive points (not negative)

We have changed the negative and positive wording throughout the manuscript.

369 – 372: why not suggest to add specifically the whole grain content as a beneficial component

see above.

422 – 435: these are important comments that limit the value of the analyses. Analyses of product compositions of the nutritionally best products on the markets would have been more meaningful.

This is indeed a limitation of our analysis. However in a future study, after the implementation of Nutri-Score in the Netherlands, we will be able to study the effects of product reformulation and Nutri-Score.

442: positive points

We have changed the negative and positive wording throughout the manuscript.

457 – 478: a certain limitation of the Nutri-Score is the reference base of 100 g that makes a product comparable within a food category but not as part of a diet. This is important for the consideration of oils for example. The limitations that were mentioned for the theoretical reformulation approaches should be mentioned here, too. In addition, for food companies it might not be the same to reformulate a product from B to reach an A but rather to avoid an E or D. It can be expected that products with different Nutri-Score ratings are not reformulated in equal amounts. Also, consumer insight is very important for these decisions. All these aspects would be expected here.

A remark is now included that Nutri-Score doesn’t include other aspects such as portion size (lines 330-333). We also discuss that our manuscript is based on a theoretical reformulation scenario analysis of a 1 point change in the Nutri-Score calculation (lines 461-475).

Round 2

Reviewer 3 Report

I went through the comments and they are fine for me.